# β-Ketophosphonates with Pentalenofuran Scaffolds Linked to the Ketone Group for the Synthesis of Prostaglandin Analogs [note 1]

**DOI:** 10.3390/ijms22136787

**Published:** 2021-06-24

**Authors:** Constantin I. Tănase, Constantin Drăghici, Miron Teodor Căproiu, Anamaria Hanganu, Gheorghe Borodi, Maria Maganu, Emese Gal, Lucia Pintilie

**Affiliations:** 1National Institute for Chemical-Pharmaceutical Research and Development-ICCF, 112 Vitan Av., 031299 Bucharest, Romania; lucia.pintilie@gmail.com; 2Organic Chemistry Center “C.D.Nenittescu”, 202B, Splaiul Independentei, 060023 Bucharest, Romania; cst_drag@yahoo.com (C.D.); dorucaproiu@gmail.com (M.T.C.); anamaria_hanganu@yahoo.com (A.H.); mmaganu@yahoo.com (M.M.); 3National Institute for R&D of Isotopic and Molecular Technologies, 67-103 Donat, 400293 Cluj-Napoca, Romania; borodi@itim-cj.ro; 4Faculty of Chemistry and Chemical Engineering, Babes-Bolyai University, Aranylános 11, 400012 Cluj-Napoca, Romania; gal.emese.81@gmail.com

**Keywords:** β-ketophosphonates, halogeno-pentalenofurane scafold, 15-substituted prostaglandin analogs, dimethyl methanephosphonate, X-ray crystallography, 15-bulky prostaglandin substituents

## Abstract

β-Ketophosphonates with pentalenofurane fragments linked to the keto group were synthesized. The bulky pentalenofurane skeleton is expected to introduce more hindrance in the prostaglandin analogues of type **III**, greater than that obtained with the bicyclo[3.3.0]oct(a)ene fragments of prostaglandin analogues **I** and **II**, to slow down (retard) the inactivation of the prostaglandin analogues by oxidation of 15α-OH to the 15-keto group via the 15-PGDH pathway. Their synthesis was performed by a sequence of three high yield reactions, starting from the pentalenofurane alcohols **2**, oxidation of alcohols to acids **3**, esterification of acids **3** to methyl esters **4** and reaction of the esters **4** with lithium salt of dimethyl methanephosphonate at low temperature. The secondary compounds **6b** and **6c** were formed in small amounts in the oxidation reactions of **2b** and **2c**, and the NMR spectroscopy showed that their structure is that of an ester of the acid with the starting alcohol. Their molecular structures were confirmed by single crystal X-ray determination method for **6c** and XRPD powder method for **6b**.

## 1. Introduction

In the prostaglandin (PG) and prostaglandin analogs (PGs), the inactivation of PGs is mainly realized by enzyme oxidation of the 15α-OH to the 15-keto group via the 15-PGDH pathway. To slow down (retard) the 15α-OH oxidation, some structural modifications have been done: the introduction of a 15-OH,15-methyl group [1], like in arbaprostil [2], prostalene [3], tioprostanide [4], etc., a 16-OH,16-methyl group at C_16_ carbon atom, like in mexiprostil [5], two methyl groups at C_16_, like in nocloprost [6], gemeprost [7,8], cyclopentyl and cyclohexyl scafolds for a few prostacyclin and carbacyclin type compounds, like: ataprost [9,10], CH-5084 [11], SC-43350 [12] for the first, and taprostene [13], U-68215 [14], RS-93427 [15] for the second. Some prostaglandins substituted at the C_16_ carbon atom with cycloalkyl, aryl, heterocyclyl [16], furyl [17], methylene-2-(2-thiophene) [18] or C_15_ with 2-indanyl [19], aryl or heteroaryl scafolds [20] are claimed in patents.

In this direction, we introduced bicyclo[3.3.0]octene or bicylo[3.3.0]octane fragments in the β-ketophosphonates used to build the ω-side chain in a selective *E*-Horner- Wadsworth -Emmons- (*E*-HWE) reaction to obtain two PG analogs of types **I** and **II** (Figure 1):

In the first type, **I**, the bicyclo[3.3.0]octene fragment is linked to the C_16_ carbon atom. This spacing from the C_15_ carbon atom by a methylene group introduces a small, but significant hindrance of the 15-PGDH enzyme to inactivate the PG analogue by oxidation of 15α-OH to the 15-keto group and keep the expected beneficial effect of the new bicyclo[3.3.0]octene fragment on the biological activity of the new PG analogs [21]. In the second type, **II**, the bulky bicyclo[3.3.0]octene and bicyclo[3.3.0]octane fragments, linked to the C_15_ carbon atom are expected to further slow down the inactivation of the PG analog by the oxidation of 15α-OH to the 15-keto group via the 15-PGDH pathway [22]. Regarding both types **I** and **II**, we hope that the newly introduced bicyclo[3.3.0]oct(a)ene fragments will have a beneficial effect on the biological activity of the new PG analogs.

In this paper, we present the synthesis of new β-ketophosphonates, **5**, with a more bulky pentalenofurane fragment, linked to the ketone group, to build the PG analogs of type **III** (Figure 2):

The biological activity of the PG analogs of type **I**–**III** and **8**–**10** was another goal we had; the synthesis of the corresponding β-ketophosphonates (used to build their ω-side chain) was only the first step for obtaining these PG analogs. As we mentioned in the previous papers [21,22], a bicyclo[3.3.0]octa(e)ne fragment is encountered in many natural products, some with high anticancer activity. As far as we know, there is no mention in the literature for a bicyclo[3.3.0]octane (octahydropentalene) or bicyclo[3.3.0]octene (hexahydropentalene) fragment being introduced in α- or ω-side chain of a PG analog. A pentalenofuran scaffold, a rigidized bicyclo[3.3.0]octane fragment with a tetrahydrofuran ring, was also never mentioned to be introduced in a PG analog. Some anticancer activity is expected to be found in the PG analogs (of type **I**–**III** and **8**–**10**) with these scaffolds in the molecule, and we will follow up on it in the next papers.

## 2. Results and Discussions

For the synthesis of new β-ketophosphonates, the key step was the reaction of an ester with the lithium salt of dimethyl methanephosphonate, like in the previous papers [21,22,23]; the β-ketophosphonates are key intermediates for building the ω-side chain in the total stereocontrolled convergent synthesis of prostaglandins and we have also used them for obtaining the intermediates used further in the hydrogenation of the double bond of the ω-side chain [24].

The synthesis of the β-ketophosphonates, presented in Scheme 1, started from the pentalenofurane alcohols **2**, previously obtained from regioselective reactions of the diol **1** [25]: by oxymercuration-demercuration reaction (compound **2a**) and respectively by haloetherification (for compounds **2b** and **2c**) [26].

Diols **2** were oxidized with 2.4 M Jones reagent to afford acids **3** in good yields. In the reaction, the secondary compounds **6b** and **6c** resulted in small quantities. Their structure was established by NMR spectroscopy which showed that they had appeared by esterification of the acid resulted in the reaction, with the starting alcohols **2b** and **2c**. This esterification is mentioned in the literature, for example in the reference [27]. A similar compound **6a** resulted also in the synthesis of **3a**, but it was not isolated pure. Next, acids **2** were esterified in good yields with methanol in the presence of TsOH as acid catalyst. In the last step, esters **4a**–**4c** were reacted with the lithium salt of dimethyl methanephosphonate at low temperature (−75 °C to −65 °C) to give the β-ketophosphonates **5a**–**5c**, also in good yield.

The β-ketophosphonates **5a**–**5c** will be used to obtain the PG analogs of type **III** (Figure 2) with the most steric hindrance to the C_15α_-OH group as compared to that encountered in compounds **I** and **II** (Figure 1); so, the inactivation of the PG analogs by oxidation of 15α-OH to the 15-keto group via the 15-PGDH pathway is expected to be even more slowed.

We used the β-ketophosphonate **5c** in the *E*-HWE selective olefination of the aldehyde **7**, with the hydrogenated α-side chain, for building the ketoprostaglandin analog **8**. As expected, the reduction of the enone group to the desired allylic alcohol **9** with the selective, but bulky reducing reagent aluminium diisobornyloxyisopropoxide, usually used in the PG field, did not proceed, as in the case of the PG analog **II** (R^1^, R^2^ = O) (Figure 1). This finding strengthens our expectation that the inactivation of the PG analogs by oxidation of 15α-OH to the 15-keto group via 15-PGDH pathway will be even more slowed than in the PG analogs of type **II**. The Luche reduction of the enone **8** with NaBH_4_ and CeCl_3_ gave the allylic alcohol **9** together with its 15-epimer, **10** (Scheme 2). So, we introduced for the first time a pentalenofurane scaffold in the ω-side chain of a F_1_ PG analog. These results will be presented separately [28].

The ^1^H- and ^13^C-NMR spectra are in agreement with the chemical structure of the compounds **2**–**6**. The analysis of the proton spectra highlights several features. All methylene groups (positions 2, 4, 7 and respectively 2′, 4′ and 8′) appear as AB systems but are highlighted as such only in the case of protons 7, 8 and 7′ and 8′. In all the other cases, the vicinal and remote couplings cause the AB system not to appear explicitly. The solving of the AB systems was done with the help of two-dimensional homonuclear and heteronuclear correlation spectra.

In the β-ketophosphonate compounds **5a**–**5c**, the H-8 protons, which form an AB system with a coupling constant of about 14 Hz, are also coupled to phosphorus with a coupling constant of 22.7–22.9 Hz. The presence of halogen in position 5 does not influence the chemical shift of H-5, but in the case of the carbon spectra of compound **5b** the strong iodine-induced shielding of C-5 is noticed, the chemical shift is lowered to δ = 32.54 ppm. β-Ketophosphonates **5a**, **5b**, and **5c** show couplings between the phosphorus atom and the methylene carbon (*J* = 128.5 Hz), respectively the C-8 carbonyl group (*J* = 6.2 Hz). The methoxy group has couplings with phosphorus (*J*_(H8-P)_ = 11.2 Hz) and carbon (*J*_(C8-P)_ = 16.7 Hz).

The synthesis of the new β-ketophosphonates **5a**–**5c** with a pentalenofuran scaffold linked to the ketone group started from the pentalenofuran alcohols **2a**–**2c**, whose chemical structure was confirmed by X-ray crystallography [26]. Chemical transformations of the exocyclic hydroxymethyl group from **2** to **5** in waiting to keep unmodified the pentalenofuran skeleton, but we tried to confirm their structure by X-ray crystallography. Though the β-ketophosphonates **5b** and **5c** were obtained crystallized as white beads and as puffy aces, the crystals were not suitable for single crystal determination of their structure, and not by XRPD powder diffraction method. The structure of the secondary compounds **6b** and **6c** was confirmed by X-ray crystallography, as follows:-Compound **6c** provided suitable crystals for single crystal X-ray determination and its structural configuration is presented in Figure 3. As we observed in NMR spectra, the molecule **6c** is an ester of the acid **3c**, formed in Jones oxidation of the alcohol **2c**, with the starting un-oxidized alcohol **2c**, presented in the reaction mixture until its complete oxidation.

-By contrast with compound **6c**, compound **6b** had no suitable single crystals and the structure was determined by the XRPD method, presented in detail at Experimental Section 3.2.

The experimental data results conducted to the structure **6b**, presented in Figure 4:

Detailed experimental procedures are presented in Experimental Section 3.2. X-ray crystallography. The main crystallographic data, bond lengths (Å) and angles (°) are given in Appendix A (ESI), and the values are in agreement with the pentalenofuran structure of both moieties of the molecules, as previously presented for pentalenofuran alcohols **2** [26].

## 3. Experimental

### 3.1. General Information

Melting points (uncorrected) were determined in open capillary on an OpiMelt melting point apparatus (MPA 100, Stanford Research System, Inc., Sunnyvale, CA, USA). The progress of the reactions was monitored by TLC on silica gel 60 or 60F_254_ plates (Merck, Darmstadt, Germany) eluted with the solvent systems: I, ethyl acetate-hexane-acetic acid, 5:1:0.1, II, ethyl acetate-hexane-acetic acid, 5:4:0.1. Spots were developed in UV and with 15% H_2_SO_4_ in MeOH (heating at 110 °C, 10 min). IR spectra were recorded on a FT-IR spectrometer (Bruker Vertex 70, Etilingen, Germany) by ATR and frequencies were expressed in cm^−1^, with the following abbreviations: w = weak, m = medium, s = strong, v = very, br = broad. HRMS spectra were recorded on LTQ Orbitrap XL (Thermo Fisher Scientific, Waltham, MA, USA). ^1^H-NMR and ^13^C-NMR spectra were recorded on a 300 MHz spectrometer (Bruker, Rheinstatten, Germany) chemical shifts are given in ppm relative to TMS as internal standard. Complementary spectra: 2D-NMR and decoupling were done for the correct assignment of NMR signals. The numbering of the atoms in the compounds is presented in Scheme 1. Compound **2a** was obtained by oxymercuration-demercuration reaction of the starting diol **1** [24], as we presented in a previous paper [25], and compounds **2b**–**2c** were obtained by haloetherification of the same diol **1** [26].

#### 3.1.1. Chemistry: Synthesis of Compound **3a**

To a solution of compound **2a** (18.09 g, 0.1075 mol) in acetone (500 mL), cooled to <−10 °C on an ice-salt bath, 2.4 M Jones reagent (85 mL) was added dropwise during 1.5 h, under energic mechanical stirring, maintaining the temperature below 0 °C. The stirring was continued for 1 h, monitoring the end of the reaction by TLC (I, R_f **2a**_ = 0.59, R_f **3a**_ = 0.76). Isopropyl alcohol (21 mL) was added, the reaction mixture was stirred for 30 min, the salts were filtered off [the salts were washed with 300 mL acetone, then dissolved in brine and extracted with ethyl acetate (2 × 150 mL)] and the acetone solutions were concentrated under reduced pressure. The concentrate was taken into the unified ethyl acetate extracts, the solution was washed with brine (50 mL), dried (MgSO_4_), concentrated to a crystallized mass (18.3 g) which was recrystallized from acetone-hexane; 10.53 g of pure (2a*S*,2a1*S*,4a*R*,5*S*,6a*R*)-octahydro-1*H*-pentaleno[1,6-bc]furan-5-carboxylic acid (**3a**) were obtained, mp 98.5–100.5 °C, FT-IR: 3375–2350 (large band, (ν_OH_), 2951vs (ν_CH2 asymm._), 2868vs (ν_CH2 symm._), 1687 (ν_C=O_), 1460s, 1425 (δ_CH2_), 1355m, 1290s (ν_C-O-C_), 1265s, 1211m, 1105m, 1075m (ν_C-OH_), 1000m, 896m (ν_C-O-C_), 801m (ν_C-O-C_), 717m, 679m, ^1^H-NMR (CDCl_3_, *δ* ppm, *J* Hz): 4.43 (dd, 1H, H-6, 3.2, 6.0), 3.74 (d, 1H, H-7, 8.9), 3.50 (dd, 1H, H-7, 4.9, 8.9), 2.98 (dt, 1H, H-6a, 6.0, 9.0), 2.88 (dq, 1H, H-3a, 6.6, 12.8), 2.73 (q, 1H, H-3, 8.0), 2.61 (dq, 1H, H-1, 4.9, 9.0), 2.06 (dt, 1H, H-4, 7.4, 12.8), 1.99 (m, 1H, H-2), 1.75 (dt, 1H, H-4, 10.4, 12.8), 1.67 (m, 1H, H-5), 1.54–1.40 (m, 2H, H-2, H-5), ^13^C-NMR(CDCl_3_, *δ* ppm): 179.24 (C-8), 85.43 (C-6), 73.25 (C-7), 54.57 (C-6a), 48.63 (C-3a), 46.46 (C-3), 44.03 (C-1), 33.46 (C-2), 32.39 (C-4), 25.99 (C-5), HRMS ESI (+) calc. for M+H, 183.1015, found [M + H] 183.1010. By LPC of the mother liquors (eluent: benzene-ethyl acetate, 2:1), another 3.32 g (total yield 81.8%) of pure compound **3a** were obtained and 2.48 g of pure starting alcohol **2a** were recovered.

#### 3.1.2. Synthesis of Compound **3b**

The iodo-pentalenofurane compound **2b** (9.76 g, 33.18 mmol) in acetone (175 mL) was oxidized under the same conditions, presented above, with 2.4M Jones reagent (29 mL), monitoring the end of the reaction by TLC (II, R_f **2b**_ = 0.48, R_f **3b**_ = 0.63. A secondary compound was formed in the reaction with R_f **6b**_ = 0.75). After the addition of isopropanol (10 mL) and work-up as previously, the concentrate was dissolved in ethyl acetate extracts (2 × 150 mL), the resulting solution was washed with brine (60 mL), 10% KHCO_3_ solution (3 × 30 mL), brine (50 mL), dried (MgSO_4_) and concentrated under reduced pressure, afording 2.8 of the neutral secondary compound (2a*R*,2a1*S*,3*R*,4a*S*,5*S*,6a*R*)-3-iodooctahydro-1*H*-pentaleno[1,6-bc]furan-5-yl)methyl (2a*R*,2a1*S*,3*R*,4a*R*,5S,6a*R*)-3-iodooctahydro-1H-pentaleno[1,6-bc]furan-5-carboxylate (**6b**), which crystallized (720 mg) from ethyl acetate-hexane, FT-IR: 2933vs (ν_CH2 asymm._), 2840vs (ν_CH2 symm._), 1700 (ν_C=O_), 1447m, 1338 (δ_CH2_), 1271s (ν_C-O-C_), 1224m, 1152s, 1153s, 992s, 936m (ν_C-C_), 908m (ν_C-O-C_), 809m (ν_C-O-C_), ^1^H-NMR (CDCl_3_, *δ* ppm, *J* Hz): 4.56 (d, 1H, H-6, 5.1), 4.55 (d, 1H, H-6′, 5.1), 4.43 (d, 2H, H-5, H-5′, 5.1), 4.04 (dd, 1H, H-8′, 6.6, 11.2), 3.94 (dd, 1H, H-8′, 8.4, 11.2), 3.81(d, 1H, H-7 or H-7′, 8.9), 3.77 (d, 1H, H-7 or H-7′, 8.9), 3.56 (d, 1H, H-7 or H-7′, 8.9), 3.53 (d, 1H, H-7 or H-7′, 8.9), 3.36–3.19 (m, 3H, H-3, H-6a, H-6′a), 3.02-2.95 (m, 2H, H-1, H-1′), 2.76–2.62 (m, 2H, H-3a, H-3′a), 2.38 (m, 1H, H-3′), 2.25–1.96 (m, 6H, 2H-4, 2H-4′, H-2, H-2′),1.88 (dd, 1H, H-4′, 10.2, 14.6),1.72 (dt, 1H, H-2, 9.7, 13.1), ^13^C-NMR (CDCl_3_, *δ* ppm): 172.77 (COO), 92.69, 92.37 (C-6, C-6′), 74.32, 74.15 (C-7, C-7′), 65.15 (C-8′), 53.41, 53.29 (C-6a, C-6a′), 46.04 (C-3), 42.31 (C-3′), 47.59 (C-3a), 46.01 (C-3a′), 43.37 (C-1), 42.31 (C-1′), 37.12 (C-4′), 35.33 (C-4), 34.59 (C-2′), 33.23 (C-2), 33.18, 32.76 (C-5, C-5′), HRMS calc. M + H: 587.0149, found: 587.2804.

The KHCO_3_ solutions were acidified to pH 2 with 2N HCl, the resulting precipitate was extracted with ethyl acetate (3 × 75 mL), the extracts were washed with brine (30 mL), dried (Na_2_SO_4_), concentrated and the crude product was crystallized from ethyl acetate-hexane, resulting in 8.5 g (85.15%) of (2a*R*,2a1*S*,3*R*,4a*R*,5S,6a*R*)-3-iodooctahydro-1*H*-pentaleno[1,6-bc]furan-5-carboxylic acid (**3b**), as prisms, mp 166.0–169.0 °C (dec), IR: 2959m, 2851m, 2641br, 1685vs, 1419m, 1249m, 1109m, 1058s, 906s, 593w, ^1^H-NMR (CDCl_3_, *δ* ppm,*J* Hz): 4.56 (d, 1H, H-6, 5.7), 4.42 (d, 1H, H-5, 3.0), 3.80 (d, 1H, H-7, 8.9), 3.54 (dd, 1H. H-7, 5.0, 8.9), 3.36–3.19 (m, 2H, H-6a, H-3a), 2.98 (dt, 1H, H-3, 7.0, 10.2), 2.68 (dq, 1H, H-1, 5.0, 9.4), 2.21 (dd, 1H, H-2, 7.4, 13.0), 2.12 (m, 2H, H-4), 1.73 (dd, 1H, H-2, 10.2, 13.0), ^13^C-NMR (CDCl_3_, *δ* ppm): 179.02 (COO), 92.76 (C-6), 74.11 (C-7), 53.35 (C-6a), 47.41 (C-3), 45.93 (C-3a), 43.40 (C-1), 37.03 (C-4), 33.47 (C-2), 32.45 (C-5), HRMS ESI (+) calc. for M + H, 308.9982, found 309.1297.

#### 3.1.3. Synthesis of Compound **3c**

Bromoalcohol **2c** (11.67 g, 47 mmol), dissolved in acetone (250 mL), was oxidized as in example 1: 2.4 M Jones reagent (37 mL), isopropyl alcohol (16 mL), TLC (II, R_f **2c**_ = 0.43, R_f **3c**_ = 0.57, R_f **6c**_ = 0.72). The work-up of the reaction as in example 2 gave 9.04 g (73.7%) of crystallized acid (2a*R*,2a1*S*,3*R*,4a*R*,5*S*,6aR)-3-bromooctahydro-1*H*-pentaleno[1,6-bc]furan-5-carboxylic acid (**3c**), mp 159–160.0 °C, IR: 2964m, 2863m, 2684br, 1690vs, 1430m, 1213m, 1065m, 910m, 620w, ^1^H-NMR (CDCl_3_, *δ* ppm, *J* Hz): 4.40 (d, 1H, H-5, 5.0), 3.78 (d, 1H, H-7, 8.9), 3.56 (dd, 1H, H-7, 5.0, 8.9), 3.28 (m, 1H, H-6a), 3.22 (m, 1H, H-3a), 2.97 (dt, 1H, H-3, 6.5, 12.8), 2.70 (dt, 1H, H-1, 5.0, 9.4), 2.18 (m, 1H, H-2), 2.14 (m, 2H, H-4), 1.73 (td, 1H, H-2, 10.2, 13.0), ^13^C-NMR (CDCl_3_, *δ* ppm): 179.01 (COO), 90.90 (C-6), 74.08 (C-7), 55.20 (C-5), 53.29 (C-6a), 47.63 (C-3), 45.10 (C-3a), 43.36 (C-1), 35.63 (C-4), 33.17 (C-2), HRMS ESI (+) calc. for M + H, 261.0120/263.0100, found 261.1088/263.0880.

By LPC purification of the ethyl acetate phases extracted with 10% KHCO_3_ soln., 1.20 g (5.5%) of pure (2a*R*,2a1*S*,3*R*,4a*S*,5*S*,6a*R*)-3-bromooctahydro-1*H*-pentaleno[1,6-bc]furan-5-yl)methyl (2a*R*,2a1*S*,3*R*,4a*R*,5*S*,6a*R*)-3-bromooctahydro-1*H*-pentaleno[1,6-bc]furan-5-carboxylate (**6c**), resulted, mp 122.0–124.0 °C, IR: 2927s, 2852s, 1711vs, 1460w, 1178vs, 1060s, 618m, ^1^H-NMR (CDCl_3_, *δ* ppm, *J* Hz): 4.46–4.37 (m, 4H, H-5, H-6′, H-5′, H-6), 4.10–3.90 (m, 2H, H-8′), 3.84–3.53 (m, 4H, 2H-7, 2H-7′), 3.35–3.15 (m, 2H, H-6a, H-6a′), 3.02–2.89 (m, 2H, H-3a, H-3a′), 2.80–2.65 (m, 2H, H-1, H-1′), 2.39 (H-3), 2.23–1.65 (m, 8H, H-3′, 2H-2, H-2′, 2H-4, 2H-4′), 1.17 (dt, 1H, H-2′, 9.9, 12.7), ^13^C-NMR (CDCl_3_, *δ* ppm): 172.85 (COO), 90.93, 90.64 (C-6, C-6′), 74.36, 74.19 (C-7, C-7′), 65.15 (C-8′), 55.78, 55.45 (C-5, C-5′), 53.47, 53.35 (C-6a, C-6a′), 47.89, 45.30 (C-3. C-3′), 44.64, 43.92 (C-3a, C-3a′), 43.45, 42.64 (C-1, C-1′), 35.80, 34.38 (C-4, C-4′), 34.05, 33.46 (C-2, C-2′), HRMS calc. M + H: 489.0271/491.0250/493.0230, found: 489.0396/491.0232/493.0210.

#### 3.1.4. Synthesis of Compound **4a**

Compound **3a** (5.26 g, 28.86 mmol) was dissolved in methanol (175 mL), TsOH (200 mg) was added and the solution was stirred overnight at rt, monitoring the end of the reaction by TLC (II, R_f **3a**_ = 0.55, R_f **4a**_ = 0.67). KHCO_3_ (0.5 g) was added, the solution was stirred for 30 min, the solvent was distilled under reduced pressure, the residue was taken in water (20 mL) and chloroform (150 mL), the phases were separated, organic phase was washed with sat. soln. NaHCO_3_ (30 mL) (aqueous phases extracted with more 30 mL chloroform), brine (30 mL), dried (Na_2_SO_4_), concentrated and the crude product was purified by LPC (ethyl acetate-hexanes, 1:1). A pure fraction of methyl (2a*S*,2a1*S*,4a*R*,5*S*,6a*R*)-octahydro-1*H*-pentaleno[1,6-bc]furan-5-carboxylate (**4a**, 4.89 g, 86.4%) was obtained as an oil, IR: 2909vs (ν_CH2 asymm._), 2833vs (ν_CH2 symm._), 1713vs (ν_C = O_), 1492 (δ_CH2_), 1347m, 1256s, 982s, 882m, ^1^H-NMR (CDCl_3_, *δ* ppm, *J* Hz): 4.21 (dd, 1H, H-6, 3.4, 6.1), 3.73 (d, 1H, H-7, 8.9), 3.65 (s, 3H, OCH_3_),3.50 (dd, 1H, H-7, 4.8, 8.9), 2.97 (dt, 1H, H-6a, 6.3, 9.3), 2.85 (dt, 1H, H-3a, 6.6, 12.8), 2.72 (m,1H, H-3), 2.61 (ddt, 1H, H-1, 4.8, 10.0, 12.9), 2.08 (m, 1H, H-4), 2.00 (m, 1H, H-2, 6.9, 12.9), 1.78 (dt, 1H, H-4, 10.2, 12.9), 1.61–1.43 (m, 2H, 2H-5), 1.45(m, 1H, H-2), ^13^C-NMR (CDCl_3_, *δ* ppm): 173.90 (COO), 85.40 (C-6), 73.38 (C-7), 54.58 (C-6a), 51.37 (CH_3_O), 48.72 (C-3a), 46.64 (C-3), 44.14 (C-1), 33.55 (C-4), 32.69 (C-2), 26.03 (C-5), HRMS ESI (+) calc. for M + H, 197.1172, found 197.1165.

#### 3.1.5. Synthesis of Compound **4b**

Compound **3b** (5.06 g, 16.4 mmol) was esterified with methanol (175 mL) and TsOH (200 mg) as acid catalyst as in example 4. TLC (II, R_f **3b**_ = 0.48, R_f **4b**_ = 0.63). The work-up as in example 4 gave 4.88 g (92.4%) of pure methyl (2a*R*,2a1*S*,3*R*,4a*R*,5*S*,6a*R*)-3-iodooctahydro-1*H*-pentaleno[1,6-bc]furan-5-carboxylate (**4b**, as long needles, mp 69.5–70.5 °C, IR: 2972s, 2945vs (ν_CH2 asymm._), 2847vs (ν_CH2 symm._), 1736vs (ν_C=O_), 1461s, 1434s (δ_CH2_), 1377s, 1271m, 1171s, 1070m, 951s, 914s (ν_C-O-C_), 810s (ν_C-O-C_), 797s, ^1^H-NMR (CDCl_3_, *δ* ppm, *J* Hz): 4.55 (d, 1H, H-6, 5.8), 4.42 (d, 1H, H-5, 5.7), 3.80 (d, 1H, H-7, 8.9), 3.66 (s, 3H, CH_3_O), 3.54 (dd, 1H, H-7, 5.0, 8.9), 3.30 (dt, 1H, H-6a, 5.8, 8.8), 3.21 (dt, 1H, H-3a, 6.8, 9.5), 2.95 (dt,1H, H-3, 6.6, 12.8), 2.68 (ddt, 1H, H-1, 5.0, 8.5, 9.9), 2.20 (dt 1H, H-2, 6.7, 13.0), 2.09 (ddt, 1H, H-4, 4.7, 9.9, 14.8), 1.99 (dd, 1H, H-4, 9.3, 14.8), 1.76 (dt, 1H, H-2, 10.0, 13.0), ^13^C-NMR (CDCl_3_, *δ* ppm): 173.42 (COO), 92.74 (C-6), 74.16 (C-7), 53.29 (C-6a), 51.57 (CH_3_O), 47.41 (C-3), 46.03 (C-3a), 43.37 (C-1), 37.04 (C-4), 33.72 (C-2), 32,82 (C-5), HRMS ESI (+) calc. for M + H, 323.0138, found 323.0128.

#### 3.1.6. Synthesis of Compound **4c**

Compound **3c** (5.41 g, 20.7 mmol) was esterified as in example 4: methanol (125 mL), TsOH (150 mg), 3 days, KHCO_3_ (400 mg), TLC (II R_f **3c**_ = 0.57, R_f **4c**_ = 0.75). The crude product was purified by LPC (eluent, ethyl acetate-hexanes, 1:2) and crystallized from ethyl acetate-hexane, resulting in 4.61 g (81.0%) of pure methyl (2a*R*,2a1*S*,3*R*,4a*R*,5*S*,6a*R*)-3-bromooctahydro-1*H*-pentaleno[1,6-bc]furan-5-carboxylate (**4c**), mp 57.5–58.0 °C, IR: 2936s, 2842s, 1723vs, 1432m, 1193m, 1163vs, 801m, 619w, ^1^H-NMR (CDCl_3_, *δ* ppm, *J* Hz): 4.40-4.39 (m, 2H, H-5, H-6), 3.78 (d, 1H, H-7, 8.8), 3.66 (s, 3H, CH_3_O), 3.56 (dd, 1H, H-7, 5.0, 8.8), 3.29 (dt, 1H, H-6a, 6.0, 8.7), 3.19 (m, 1H, H-3a), 2.94 (dt,1H, H-3, 6.6, 12.9), 2.69 (dq, 1H, H-1, 5.0, 9.4), 2.21-2.09 (m, 2H, H-2, H-4), 2.01 (dd, 1H, H-4, 9.3, 14.1), 1.75 (dt, 1H, H-2, 10.0, 13.0), ^13^C-NMR (CDCl_3_, *δ* ppm): 173.41 (COO), 90.93 (C-6), 74.17 (C-7), 55.47 (C-5), 53.27 (C-6a), 51.55 (CH_3_O), 47.68 (C-3), 45.25 (C-3a), 43.39 (C-1), 35.64 (C-4), 33.42 (C-2), HRMS ESI (+) calc. for M + H, 275.0277/277.0256, found 275.0269/277.0248.

#### 3.1.7. Synthesis of β-Ketophosphonate **5a**

To a solution of 93.6% dimethyl methanephosphonate (7.42 g 100%, 56 mmol) in anhydrous tetrahydrofuran (THF) (60 mL), cooled to −70 °C under an argon atmosphere using an ULTRA-KRYOMAT system (Lauda Dr. R. Wobser & Co, Germany, a solution of 1.5 M *n*-butyllitium (39 mL) in hexane was added dropwise under mechanical stirring, maintaining the reaction temperature below −65 °C. The stirring was continued for 30 min, a solution of the ester compound **4a** (4.21 g, 21.45 mmol) in THF (25 mL) was added dropwise and the progress of the reaction was followed by TLC (II, R_f **4a**_ = 0.67, R_f **5a**_ = 0.16). Acetic acid (3.7 mL) was added, stirring was continued for 15 min, the reaction mixture was transferred into a round bottom flask and concentrated under reduced pressure. The residue was taken in water (70 mL) and chloroform (100 mL), the phases were separated (aqueous phase was extracted with 2 × 100 mL chloroform), the organic phase was washed with brine (70 mL), dried (Na_2_SO_4_) and concentrated. The crude product (11.7 g) was purified by LPC (ethyl acetate-hexanes, 1:1), resulting in 4.10 g (88.0%) of the pure β-ketophosphonate dimethyl (2-((2a*S*,2a1*S*,4a*R*,5*S*,6a*R*)-octahydro-1*H*-pentaleno[1,6-bc]furan-5-yl)-2-oxoethyl)phosphonate (**5a**), as an oil, ^1^H-NMR (CDCl_3_, *δ* ppm, *J* Hz): 4.21 (dd, 1H, H-6, 3.7, 6.3), 3.80 (d, 3H, CH_3_OP, 11.3), 3.79 (d, 3H, CH_3_OP, 11.3), 3.73 (d, 1H, H-7, 9.0), 3.51 (dd, 1H, H-7, 4.9, 9.0), 3.26 (dd, 1H, CH_2_P, *J*_H-P_ = 22.7, *J*_vic_ = 14.0), 3.258 (dt, 1H, H-3, 6.7, 12.4), 3.05 (dt, 1H, H-6a, 6.4, 9.3), 3.01 (dd, 1H, CH_2_P, *J*_H-P_ = 22.7, *J*_vic_ = 14.0), 2.79 (dt, 1H, H-3a, 4.9, 10.2), 2.65 (ddt, 1H, H-1, 4.8, 8.5, 12.6), 1.99 (m, 2H, H-2, H-4), 1.79 (dt, 1H, H-2, 10.2, 12.6), 1.50 (m, 1H, H-4), 1.45 (m,1H, H-5), 1.38 (dt, 1H, H-5, 3.7, 6.3), ^13^C-NMR (CDCl_3_, *δ* ppm, *J* Hz): 201.58 (CO, *J_C-P_* = 6.2), 85.02 (C-6), 73.42 (C-7), 57.79 (C-6a), 55.04 (C-3), 53.22 (d, CH_3_OP, *J*_C-P_ = 6.7), 53.02 (d, CH_3_OP, *J*_C-P_ = 6.7), 46.44 (C-3a), 44.01 (C-1), 40.50 (d, CH_2_P, *J*_C-P_ = 129.4), 37.72 (C-4), 31.50 (C-2), 25.68 (C-5).

#### 3.1.8. Synthesis of β-Ketophosphonate **5b**

The β-ketophosphonate **5b** was synthesized as in example 7: 93.6% dimethyl methanephosphonate (4.66 g 100%, 37.6 mmol) in anhydrous THF (35 mL), 1.5 M *n*-butyllitium (24.5 mL) in hexane, ester **4b** (4.64 g, 14.4 mmol) in THF (20 mL), acetic acid (2.5 mL), TLC (II, R_f **4b**_ = 0.77, R_f **5b**_ = 0.18). The crude product was crystallized from ethyl acetate-hexane, resulting 4.69 g (78.6%) of pure β-ketophosphonate dimethyl (2-((2a*R*, 2a1*S*,3*R*,4a*R*,5*S*,6a*R*)-3-iodooctahydro-1*H*-pentaleno[1,6-bc]furan-5-yl)-2-oxoethyl)phosphonate (**5b**), as white beads, mp 70.5–72.5 °C, IR: 2951s, 2850m, 1696s, 1460m, 1245s, 1167m, 1024vs, 957m, 806s, 547w, 505m, ^1^H-NMR (CDCl_3_, *δ* ppm, *J* Hz): 4.49 (d, 1H, H-6, 5.7), 4.36 (d, 1H, H-5, 4.5), 3.80 (d, 3H, CH_3_OP, 11.1), 3.79 (d, 3H, CH_3_OP, 11.1), 3.76 (d, 1H, H-7, 8.9), 3.50 (dd, 1H, H-7, 5.0, 8.9), 3.23-3.12 (m, 3H, H-6a, H-3a, H-3), 3.18 (dd, 1H, CH_2_P, *J*_H-P_ = 22.9, *J*_vic_ = 13.7), 2.97 (dd, 1H, CH_2_P, *J*_H-P_ = 22.9, *J*_vic_ = 13.7), 2.67 (dq, 1H, H-1, 5.3, 9.4), 2.06 (dd, 1H, H-2, 6.4, 10.4), 1.98-1.89 (m, 2H, H-4), 1.75 (dt, 1H, H-2, 10.4, 13.8), ^13^C-NMR (CDCl_3_, *δ* ppm): 201.58 (CO, *J_C-P_* = 6.2), 92.39 (C-6), 74.05 (C-7), 56.11 (C-3), 53.59 (C-6a), 53.08 (d, CH_3_OP, *J*_C-P_ = 6.6), 53.00 (d, CH_3_OP, *J*_C-P_ = 6.6), 45.74 (C-3a), 43.16 (C-1), 40.80 (CH_2_P, *J*_C-P_ = 128.3), 36.49 (C-4), 32.63 (C-2), 32.54 (C-5), HRMS, calc.: [M + 1] 415.01658, found: 415.01584.

#### 3.1.9. Synthesis of β-Ketophosphonate **5c**

The β-ketophosphonate **5c** was synthesized as in example 7: 93.6% dimethyl methanephosphonate (5.03 g 100%, 40.5 mmol, 5.4 mL) in anhydrous THF (50 mL), 1.5 M *n*-butyllitium (27 mL) in hexane, bromo-ester **4c** (4.46 g, 16.2 mmol) in THF (25 mL), acetic acid (3 mL), TLC (II, R_f **4c**_ = 0.75, R_f **5c**_ = 0.21). The crude product was crystallized from ethyl acetate-hexane, resulting 4.96 g (83.3%) of pure β-ketophosphonate dimethyl (2-((2a*R*,2a1*S*,3*R*,4a*R*,5*S*,6a*R*)-3-bromooctahydro-1*H*-pentaleno[1,6-bc]furan-5-yl)-2-oxo-ethyl)phosphonate (**5c**), as puffy needless, mp 75.5–77.0 °C, IR: 2948m, 2856m, 1696s, 1289m, 1211s, 1186m, 1019vs, 807s, 616w, 561m, ^1^H-NMR (CDCl_3_, *δ* ppm, *J* Hz): 4.36 (m, 2H, H-6, H-5), 3.79 (d, 3H, CH_3_OP, 11.2), 3.75 (d, 3H, CH_3_OP, 11.2), 3.73 (d, 1H, H-7, 8.9), 3.54 (dd, 1H, H-7, 5.1, 8.9), 3.36-3.18 (m, 3H, H-6a, H-3a, H-3), 3.19 (dd, 1H, CH_2_P, *J*_H-P_ = 22.6, *J*_vic_ = 13.7), 2.97 (dd, 1H, CH_2_P, *J*_H-P_ = 22.6, *J*_vic_ = 13.7), 2.69 (dq, 1H, H-1, 5.3, 9.4), 2.09-1.95 (m, 2H, H-2, H-4), 1.87 (dd, 1H, H-4, 7.0, 14.4), 1.74 (dt, 1H, H-2, 10.4, 12.9), ^13^C-NMR (CDCl_3_, *δ* ppm): 201.14 (CO, *J*_C-P_
*=* 3.0), 90.47 (C-6), 74.09 (C-7), 53.60 (C-6a), 55.34 (C-5), 56.18 (C-3), 53.10 (d, CH_3_OP, *J*_C-P_ = 6.1), 53.00 (d, CH_3_OP, *J*_C-P_ = 6.1), 44.97 (C-3a), 43.20 (C-1), 40.50 (CH_2_P, *J*_C-P_ = 128.4), 35.13 (C-4), 32.25 (C-2), HRMS calc. [M + 1] for both isotopes: 367.03045 and 369.02840, found: 367.02904 and 369.02672.

### 3.2. X-ray Crystallography

A suitable single crystal sample of **6c** was mounted on a goniometer of the SuperNova diffractometer (Rigaku Corporation, Tokyo, Japan) equipped with dual micro-sources (Mo and Cu) and a CCD detector (Eos, Rigaku Corporation, Tokyo, Japan) experimental data being collected with CuKα radiation, X-ray tube set at 50 kV and 0.8 mA. Data collection strategy, Lorentz-polarization and absorption corrections was conducted with CrysAlis PRO [29] and absorption correction was done using the empirical Multi-scan method which uses spherical harmonics in SCALE3 abs pack algorithm. Crystal structure was solved by direct methods using SHELXS [30] and refined by least square minimization using ShelXL [31], both these programs being implemented in Olex2 (v.21.3) [32] software. All non-hydrogenoid atoms were localized by Fourier difference map and refined anisotropically with the displacement isotropic parameter Uiso(H) = 1.2Ueq(C) for all CH and CH_2_ groups. Hydrogen atoms were placed in idealized positions and treated as riding as follows: ternary CH refined with riding coordinates (C-H = 0.98 Å) and secondary CH_2_ refined with riding coordinates (C-H = 0.97 Å). By Fourier maps an unassigned electron density is observed around the second site for the carbonyl atom. This disorder was modeled in the sense that the O1 oxygen of the carbonyl group was located at two positions with occupancy factors 0.69 and 0.31, but O1 is linked chemically only to C8 or C8′.

For compound **6b** it was not possible to obtain suitable single crystals for the determination of the crystal structure, so an attempt was made to solve the crystal structure by the XRPD method. The methodology applied for determining this structure is presented in ESI (Section 4).

The molecular configuration of the molecules **6c** and **6b** are presented in Figure 3 and Figure 4 in Results and Discussion, and represent asymmetric units, both **6c** and **6b** having a single molecule in the asymmetric unit. The two molecular configurations are similar but not insofar as they can overlap. Each of the two molecules (**6c** and **6b**) has six rings, every ring has five members (atoms). These rings adopt approximately a distorted envelope conformation. An ideal envelope conformation has a plane of symmetry that passes through one of the five ring atoms and is perpendicular to the opposite side. In order to estimate the deformation degree of the five member rings related to existence of the symmetry plane, ΔCs [33] was defined with the relation and figure below (Figure 5), in which Φ_i_ are torsion angles of each ring. Based on the relation below, ΔCs was calculated for each atom of the rings and those ΔCs with the lowest value were chosen. The results are shown in Table 1. It can be seen that the planes of symmetry pass through the same atoms for molecules **6c** and **6b** with one exception for each molecule: in molecule **6c** the symmetry plane passes through C7′ and in molecule **6b** the symmetry plane passes through O7′.
$$\Delta C_{s}=\sqrt{\frac{\sum_{i=1}^{2}\left(\Phi_{i}+\Phi_{i}^{\prime }\right)}{2}}$$

It is worth mentioning that compound **6c** crystallized in the triclinic system P-1 space group with two molecules in the unit cell, having the following lattice parameters: a = 8.1233 Å, b = 9.8197 Å, c = 12.7861 Å, α = 97.873°, β = 100.177°, γ = 95.683°. Compound **6b** belongs to a monoclinic system P21/n space group with four molecules in the unit cell, having the following lattice parameters: a = 5.5121 Å, b = 11.2583 Å, c = 32.2033 Å, α = 90˚, β = 94.527°, γ = 90°.

CCDC-2068869 (for **6c**) and CCDC-2068870 (for **6b**) contains the crystallographic data for this contribution. For the compound 6c, the prime symbol was replaced with P, for example C1′ was replaced with C1P. These data can be obtained free of charge via www.ccdc.cam.ac.uk/conts/retrieving.html deposited at 22 June 2021 (or from the Cambridge Crystallographic Data Centre, 12 Union Road, Cambridge CB2 1EZ, UK; fax: (+44)-1223-336-033; or deposit@ccdc.ca.ac.uk).

## 4. Conclusions

β-Ketophosphonates with pentalenofurane fragments linked to the keto group were synthesized. Their synthesis started from the pentalenofurane diols **2** and was realized following a sequence of three high yielding steps. Small quantities of secondary compounds **6b** and **6c** were isolated pure and the NMR spectroscopy showed that their structures are esters of the newly formed acids **3b** and **3c** with the starting diols **2b** and **2c**. The molecular structures of secondary compounds **6c** and **6b** was confirmed by X-ray crystallography, using two different methods: single crystal X-ray determination method for **6c** and XRPD powder method for **6b**. The β-ketophosphonates **5a**–**5c** will be used as key intermediates for building the ω-side chain of prostaglandins of type **III**. β-Ketophosphonate **5c** was already used for obtaining the first PGF_1_ analogs containing this pentalenofurane scaffold in the ω-side chain of the molecule [28].

## 5. Patents

A patent application [34] describing the synthesis of β-ketophosphonates having a functionalized pentalenofurane fragment has been submitted to OSIM, Romania.

## Data Availability

Not applicable.

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
