# Peer review of "β-Ketophosphonates with Pentalenofuran Scaffolds Linked to the Ketone Group for the Synthesis of Prostaglandin Analogs†"

_ijms, 2021, doi:10.3390/ijms22136787_

Round 1
Reviewer 1 Report
In this manuscript by Tănase et al. the authors describe the synthesis of prostaglandin analogs that are purported to undergo hydrolysis at a much lower rate of oxidation at the 15alpha-OH site. Overall, the manuscript is clearly written, but focuses so intently on the step-by-step synthesis that the importance is somewhat lost. The authors go into sufficient detail regarding the synthesis of these compounds and that portion of the manuscript is relatively easy to follow. However, there are some major shortcomings in this manuscript as presented. First, there is almost no background provided into prostaglandin biological action or the therapeutic utility of the synthesized compounds. The authors instead launch directly into the description of the production of a series of compounds without stressing their importance. Indeed, given the potential therapeutic impact the introduction and abstract steer directly to the organic synthesis, but spend little time placing anything within a useful context.
While the organic synthesis itself is clearly described, the structural data are all hidden in the SI. A portion of the NMR data, at least, should find its way into the main text. This is especially true as the relevant chemical shifts are provided within the text of the experimental section.
While I recognize this is largely an organic synthesis paper, it would be useful if the authors could also provide evidence for the improved oxidation properties of the synthesized compounds. As these compounds appear to have a pending patent I assume some sort of data may be available.
Author Response
Thank you very much for your valuable observations.
We revised the manuscript according with your recommendations, as follows:
Comments and Suggestions for Authors:
- In this manuscript by Tănase et al. the authors describe the synthesis of prostaglandin analogs that are purported to undergo hydrolysis at a much lower rate of oxidation at the 15alpha-OH site. Overall, the manuscript is clearly written, but focuses so intently on the step-by-step synthesis that the importance is somewhat lost. The authors go into sufficient detail regarding the synthesis of these compounds and that portion of the manuscript is relatively easy to follow. However, there are some major shortcomings in this manuscript as presented. First, there is almost no background provided into prostaglandin biological action or the therapeutic utility of the synthesized compounds. The authors instead launch directly into the description of the production of a series of compounds without stressing their importance. Indeed, given the potential therapeutic impact the introduction and abstract steer directly to the organic synthesis, but spend little time placing anything within a useful context.
-We introduced the following paragraph before Results and discussions: The biological activity of the PG analogs of type I-III and 8-10 was another goal we had; the synthesis of the corresponding β-ketophosphonates (used to build their w-side chain) was only the first step for obtaining the PG analogs. As we mentioned in the previous papers [21, 22], a bicyclo[3.3.0]octa(e)ne fragment is encountered in many natural products, between them with high anticancer activity. As far as we know, there is no mention in the literature for a bicyclo[3.3.0]octane (octahydropentalene) or bicyclo[3.3.0]octene (hexahydropentalene) fragment to be introduced in the α- or w-side chain of a PG analog. A pentalenofurane scaffold, a rigidized bicyclo[3.3.0]octane fragment with a tetrahydrofurane ring, was also never mentioned to be introduced in a PG analog. Some anticancer activity is expected to be found in the PG analogs (of type I-III and 8-10) with these scaffolds in the molecule, and we will follow up on it in the next papers.
- While the organic synthesis itself is clearly described, the structural data are all hidden in the SI. A portion of the NMR data, at least, should find its way into the main text. This is especially true as the relevant chemical shifts are provided within the text of the experimental section.
We introduced a paragraph in the Results and discussions, as follow:
-The 1H- and 13C-NMR spectra are in agreement with the chemical structures of the compounds 2-6. The analysis of the proton spectra highlights several features. All methylene groups (positions 2, 4, 7 and respectively 2 ', 4' and 8') appear as AB systems, but are highlighted as such only in the case of protons 7, 8 and 7' and 8 '. In all the other cases, the vicinal and remote couplings cause the AB system not to appear explicitly. The solving of the AB systems was done with the help of two-dimensional homonuclear and heteronuclear correlation spectra.
In the β-ketophosphonate compounds 5a-5c, the H-8 protons, which form an AB system with a coupling constant of about 14 Hz, are also coupled to phosphorus with a coupling constant of 22.7-22.9 Hz. The presence of halogen in position 5 does not influence the chemical shift of H-5, but in the case of the carbon spectra of compound 5b the strong iodine-induced shielding of C-5 is noticed, the chemical shift is lowered to δ = 32.54 ppm. β-Ketophosphonates 5a, 5b, and 5c show couplings between the phosphorus atom and the methylene carbon (J = 128.5 Hz), respectively the C-8 carbonyl group (J = 6.2 Hz). The methoxy group has couplings with phosphorus (J(H8-P) = 11.2 Hz) and carbon (J(C8-P) = 16.7 Hz).
- While I recognize this is largely an organic synthesis paper, it would be useful if the authors could also provide evidence for the improved oxidation properties of the synthesized compounds. As these compounds appear to have a pending patent I assume some sort of data may be available.
-We have not studied yet the chemical oxidation of the PGF1 analogs, 9 and 10, to mimic the enzyme oxidation to the 15-keto-PGF1 8. In fact, the oxidation reagents are small molecules that could oxidize the 15-OH to 15-keto group more quickly than the bulky enzyme does. Thank you for this proposal, we will keep it in mind for the next papers.
Reviewer 2 Report
Tanase et al. report on beta-ketophosphonates with pentalenofurane moieties bound to the keto group. The manuscript might be suitable for publication in International Journal of Molecular Sciences after a thorough revision.
- The Title is too long. The "3." remains unclear. If it indicates that the article is the third one of a series, this should be stated in a title footnote or so. "New" should be omitted. The same holds for "X-ray crystallography".
- Line 22: a space is missing between "the" and "5-PGDH".
- Line 29: the last sentence of the abstract may be omitted, because this is standard.
- Line 45: "thiofen" should be "thiophene".
- Figures 1 and 2 may be schemes rather then figures actually. The two captions in each case are strange. Please label the formulae with (a) and (b) or I and II or so and rewrite the captions in a way like "Scheme 1 (a).... (b) ...."
- Line 59: It is rather unnecessary to start a new paragraph here.
- Line 86: Statements like "These compounds were isolated pure by LPC and their structure was established by NMR spectroscopy" can probably omitted, as this is standard.
- Line 104: omit "in the meantime", and "HEW" should be "HWE".
- Scheme 1: compound 1 could be omitted, since the conversion of 1 to 2 is obviously not a part of the present work.
- Line 109: "strength" should be "strengthens" here.
- The section heading "X-ray crystallography" should be removed and the content partially moved to the experimental section (e.g. CCDC deposition statement), the supporting information or placed in a footnote and the remainders shoukd be merged with the "Results and discussion" section. Moreover, only side products ("secondary products", as you say) were characterized by X-ray crystallography. These should not be central to the work.
- Experimental section: the listings of spectroscopic data should be consistent throughout. NMR coupling constants may be given as "J = 5.0 Hz" or so. 1H and 13C NMR and 2D spectra (when used for signal assignments) for new compounds must the depicted in the supporting It is not necessary to list all IR bands. Those that can be assigned to characteristic vibrations should suffice. It looks as if HRMS data are missing for some compounds.
- Line 382: "The suitable single crystal..." should be "A suitable single crystal...".
- Subsection 4.2 should be significantly shorted. Especially, the PXRD and Rietveld part should be moved to the supporting information, as it has really no implications for the purpose of the article.
- Conclusions: this section could be more conclusive and less like an abstract. The first paragraph of this section is totally informative and the last sentence of this section can safely be omitted.
- Line 555: "P.R.O. CrysAlis" should probably be "CrysAlisPro".
- Line 556: The 2008 SHELX reference can be removed. For SHELXT cite ref. 31 and for SHELXL please cite G. M. Sheldricks 2015 Acta Cryst. C. article.
- Line 565: The journal name (i.e. J. Appl. Cryst.) is missing here.
There are issues with the crystal structure refinement of compound 6c:
- Inspection of the difference electron density map reveals that positional disorder of the carbobyl group has most likely been overlooked (see screenshot attached). Q1 (2.45 eA-3, green mesh) should be the second site for the carbonyl atom. Moreover, the negative difference density (red mesh) at O1 indicates that too much electron density was assigned to this site. The disorder should be modelled properly.
- No need to apply DFIX restraints here. In the disorder model, SADI restraints should be appropriate.
- Primed atom labels should only be used for the disorder part with minor occupancy.
- The atom list should be sorted.
- Extracting the HKL and INS files from the CIF results in a checksum error (also revealed by checkCIF). This could be a problem of Olex2 and is easy to avoid by generating the final CIF directly with SHELXL from the command line or so.

Author Response
Thank you very much for your valuable observations.
We revised the manuscript in agreement with your observations, as follows:
- The Title is too long. The "3." remains unclear. If it indicates that the article is the third one of a series, this should be stated in a title footnote or so. "New" should be omitted. The same holds for "X-ray crystallography".
- We changed the title to: β-Ketophosphonates with pentalenofurane scaffolds linked to the ketone group for the synthesis of prostaglandin analogs.
- Line 22: a space is missing between "the" and "5-PGDH". Resolved
- Line 29: the last sentence of the abstract may be omitted, because this is standard. Resolved
- Line 45: "thiofen" should be "thiophene". Resolved
- Figures 1 and 2 may be schemes rather then figures actually. The two captions in each case are strange. Please label the formulae with (a) and (b) or I and II or so and rewrite the captions in a way like "Scheme 1 (a).... (b) ...
- Our proposal is that we keep Figure 1 as a figure. The two chemical (structural) formulas in Figure 1 are labeled as I and II . We changed the caption of Figure 1 to “Prostaglandin analogs of type I with a bicyclo[3.3.0]octene fragment, and respectively of type II with bicyclo[3.3.0]octene and bicyclo[3.3.0]octane fragments in the ω-side chain”.
- Figure 2 seems correct in our opinion.
- Line 59: It is rather unnecessary to start a new paragraph here.
- Line 86: Statements like "These compounds were isolated pure by LPC and their structure was established by NMR spectroscopy" can probably omitted, as this is standard.
- The phrase was modified as follows: Their structure was established by NMR spectroscopy which showed that they had appeared by esterification of the acid resulted in the reaction, with the starting alcohols 2b and 2c.
- Line 104: omit "in the meantime", and "HEW" should be "HWE". “In the meantime” was removed. HEW is correct. In the manuscript we completed the phrase at line 48 with “in a selective E-Horner-Emmons-Wadsworth (E-HEW) reaction”
- Scheme 1: compound 1 could be omitted, since the conversion of 1 to 2 is obviously not a part of the present work.
-Our opinion is that compound 1 should remain in Scheme 1, because: we discuss compound 1 in the text as the starting point for compounds 2. It is more useful for readers to find directly the information for obtaining compounds 2 from compound 1, clearly outlined in Scheme 1. The removal of compound 1 would also require changing the numbering of all compounds, not only in the Scheme, but also in the rest of the manuscript (including the experimental part and the Supplementary Material).
- Line 109: "strength" should be "strengthens" here. Resolved
- The section heading "X-ray crystallography" should be removed and the content partially moved to the experimental section (e.g. CCDC deposition statement), the supporting information or placed in a footnote and the remainders shoukd be merged with the "Results and discussion" section. Moreover, only side products ("secondary products", as you say) were characterized by X-ray crystallography. These should not be central to the work.
-The section "X-ray crystallography" was removed, some phrases were also removed: The solving a structure by XRPD, briefly consists in the following steps: powder diffraction indexing, structural model obtaining and Rietveld refinement. The compound 6c crystallized in the triclinic system, P-1 space group with two molecules in the unit cell, having the following lattice parameters: a=8.1233 Å, b=9.8197 Å, c=12.7861 Å, α=97.873˚, β=100.177˚, g=95.683˚. The compound 6b belongs to P21/n space group of the monoclinic system with four molecules in the unit cell, having the following lattice parameters: a=5.5121 Å, b=11.2583 Å, c=32.2033 Å, α=90˚, β=94.527˚, g=90˚. For both compounds 6c and 6b there is a single molecule in the asymmetric unit. The two molecular configurations are similar, but not insofar as they can overlap, mainly due to the contribution of the halogen atoms with different volumes and crystallization in two different crystallographic systems. Both 6c and 6b contain five membered rings with various distorted envelope conformations (See Experimental section, 3 4.2 for detailed conformational characterization and Table S1 in Electronic Supplementary Information (ESI)),
and which was successfully solved by X-ray single crystal diffraction, from the first line.
-CCDC deposition statement was moved to the Experimental section.
- Experimental section: the listings of spectroscopic data should be consistent throughout. NMR coupling constants may be given as "J = 5.0 Hz" or so. 1H and 13C NMR and 2D spectra (when used for signal assignments) for new compounds must the depicted in the supporting. It is not necessary to list all IR bands. Those that can be assigned to characteristic vibrations should suffice. It looks as if HRMS data are missing for some compounds.
The answers for this section, are: The listings of spectroscopic data are a little corrected and are consistent throughout. Copies of 1H-, 13C- and 2D-NMR spectra for all compounds are given in the Electronic Supplementary Information (ESI). IR spectra were reduced to the characteristic vibration bands of each compound. HRMS missing data for the compounds have been done, with the exception of the compound 5a, from which we have no more amount. Copies of HRMS spectra are given in ESI.
- Line 382: "The suitable single crystal..." should be "A suitable single crystal...". Resolved
- Subsection 4.2 should be significantly shorted. Especially, the PXRD and Rietveld part should be moved to the supporting information, as it has really no implications for the purpose of the article.
Answer: The text in blue was moved in ESI.
- Conclusions: this section could be more conclusive and less like an abstract. The first paragraph of this section is totally informative and the last sentence of this section can safely be omitted.
-The last sentence: The compounds were characterized by IR, high resolution 1H- and 13C-NMR spectroscopy, MS spectroscopy and X-ray crystallography for two secondary compounds. was removed.
-In the second paragraph, the sentence: in which the bulky pentalenofurane fragment is expected to slow down (retard) the inactivation of the PG analogs by oxidation of 15α-OH to the 15-keto group via the 15-PGDH pathway was removed.
- Line 555: "P.R.O. CrysAlis" should probably be "CrysAlisPro".
Answer: This was corrected
- Line 556: The 2008 SHELX reference can be removed. For SHELXT cite ref. 31 and for SHELXL please cite G. M. Sheldricks 2015 Acta Cryst. C. article.
Answer: References were corrected
- Line 565: The journal name (i.e. J. Appl. Cryst.) is missing here. Resolved
- There are issues with the crystal structure refinement of compound 6c:
-Inspection of the difference electron density map reveals that positional disorder of the carbobyl group has most likely been overlooked (see screenshot attached). Q1 (2.45 eA-3, green mesh) should be the second site for the carbonyl atom. Moreover, the negative difference density (red mesh) at O1 indicates that too much electron density was assigned to this site. The disorder should be modelled properly.
Answer: Indeed, from Fourier maps an unassigned electron density is observed around the second site for the carbonyl atom. This disorder was modeled in the sense that the O1 oxygen of the carbonyl group was located at two positions with occupancy factors 0.69 and 0.31, but O1 is linked chemically only to C8 or C8’.
-No need to apply DFIX restraints here. In the disorder model, SADI restraints should be appropriate.
Answer: in the new version of the CIF file, the DFIX command was not used
-Primed atom labels should only be used for the disorder part with minor occupancy.
Answer: Since the molecule is symmetrical and to be consistent with Scheme 1 were denoted by prime.
-The atom list should be sorted.
Extracting the HKL and INS files from the CIF results in a checksum error (also revealed by checkCIF). This could be a problem of Olex2 and is easy to avoid by generating the final CIF directly with SHELXL from the command line or so
Answer: The new version of CHECK CIF does not present A and B alerts
Reviewer 3 Report
This manuscript, written by Tanase et al., focuses on beta-ketophosphonates with pentalenofurane moieties bound to the keto group. Authors confirmed the molecular structures of these compounds.
The manuscript, in general, is well-written. I suggest the acceptance of this manuscript in its present form.
Author Response
Thank you very much for reviewing and suggesting the acceptance of our manuscript for publication.
Round 2
Reviewer 1 Report
Accept in present form.
Author Response
Thank you very much for carefully reviewer of our manuscript.
Reviewer 2 Report
The authors have revised the manuscript. Not all points from the last round of review have been addressed. There are still issues that need to be resolved before publication can commence.
- Obviously, something with the title footnote went technically wrong.
- Still, I find the multiple captions to Figure 1 and 2 odd and unnecessary.
- I honestly do not see the need to include the paragraph starting in line 175 in the main text of the article. These crystallographic data have no implications for the results and discussion part and are never referred to. These should be moved to the experimental part or placed in a note.
- Not to provide HRMS data for one newly reported compound, because you do not have the sample anymore seems to be an odd excuse in scientific publishing. Such is not scientific rigour. Minor point: it is sometimes "HR-MS" and "HRMS" and "calctd" and "calc.". Such inconsistencies should be eliminated.
- It is perhaps confusing that the moleculare structure of 6c is shown before that of 6b and not vice versa.
- Interesting that you think that "HEW" is "correct". A bit of googling for "E-HEW" merily reveals a couple of hits leading to previous papers by the same authors. I think the reaction is commonly known as Horner-Wadsworth-Emmons (HWE): https://en.wikipedia.org/wiki/Horner%E2%80%93Wadsworth%E2%80%93Emmons_reaction Not the other way round.
- I strongly recommend moving Figure 5 to the ESI.
X-ray crystallography:
- First of all, the bad checksum issue has not been fixed. If you do a full checkCIF, you will encouter the following:
================================================================================
ALERT_012 Type_1 CIF Construction/Syntax Error, Inconsistent or Missing Data.
================================================================================
The supplied CIF contains a '_shelx_res_file' record but not a valid
associated '_shelx_res_checksum' record. A valid pair of embedded .res and
.hkl files allows the automatic creation of a .fcf file with SHELXL20xy to be
used for a detailed analysis of the refinement result. A file is valid when
the calculated and reported checksums are identical. Only characters with an
ASCII value higher than 32 contribute to the checksum.
An embedded .res file might be broken, either due to tranfer errors or to
deliberate or accidental post-refinement editing of its content.
- The disorder model is insuffcient, because the methylene H atoms at C8 and C8' are missing. As a result the sum formula should also be incorrect. The most common way to add these H atoms in SHELXL would be to split the C8 and C8' atoms, apply EXYZ and EADP constrains, assign the respective free variable for the occupacies and use appropriate PART instructions. Then, the H atoms can be added using HFIX 23.
- Still I think that primed atom labels are commonly used for disorder in X-ray crystalography.
- Please do sort the atom list in the SHELXL instruction file. Such a random atom list in the RES and CIF file is a nuisance.
- Displacement ellipsoids should be depicted for the atoms in the structure, not merily a ball-and-stick model. Structure pictures need to be updated of course, when the structure refinement was modified.
Author Response
Dear Reviewer,

Round 3
Reviewer 2 Report
The changes suggested in the last round of review have mostly been applied.
I have checked the deposited CIF for 6c again. Maybe it has been forgotten to update it at the CCDC? Likewise, the structure pictures in the article have not been updated. There appears to be an old one with two missing H atoms (those at C8) and unsorted atom list.
Regarding missing HRMS data for one compound reported, I do not know what the policy of the journal is.
Finally, please make sure that all atom labels throughout in the manuscript, SI, figures and tables correspond to those in the final CIF.
Author Response
Thank you for your observations.
We followed your recommendations, as presented below:
- I have checked the deposited CIF for 6c again. Maybe it has been forgotten to update it at the CCDC? Likewise, the structure pictures in the article have not been updated. There appears to be an old one with two missing H atoms (those at C8) and unsorted atom list.
We changed the deposited CIF for the compound 6c
- Finally, please make sure that all atom labels throughout in the manuscript, SI, figures and tables correspond to those in the final CIF.
We changed the numbering of the atoms in the Checkcif, as presented in the following Figure:
The numbering of all atoms is the same in the manuscript, ESI, figures and tables. For the Checkcif of the compound 6c we introduced the following phrase compound 6c, the prime symbol was replaced with P, for example C1’ was replaced with C1P, after the Phrase: CCDC-2068869 (for 6c) and CCDC-2068870 (for 6b) contains the crystallographic data for this contribution.
I hope You are agree with our answer.
We thank again.
Yours sincerely,
Constantin Tanase